# Effects of *Bacillus thuringiensis* HC-2 Combined with Biochar on the Growth and Cd and Pb Accumulation of Radish in a Heavy Metal-Contaminated Farmland under Field Conditions

**DOI:** 10.3390/ijerph16193676

**Published:** 2019-09-30

**Authors:** Zigang Li, Peng Wang, Xiaoyu Yue, Jingtao Wang, Baozeng Ren, Lingbo Qu, Hui Han

**Affiliations:** 1School of Food and Bioengineering, Henan University of Animal Husbandry and Economy, Zhengzhou 450046, China; zigangli@hnuahe.edu.cn (Z.L.); zzmz_w@126.com (P.W.); yuerain@163.com (X.Y.); 2State Key Laboratory of Motor Vehicle Biofuel Technology, Nanyang 473000, China; jingtaowang@zzu.edu.cn (J.W.); renbz@zzu.edu.cn (B.R.); qlb@zzu.edu.cn (L.Q.); 3School of Chemical Engineering, Zhengzhou University, Zhengzhou 450001, China; 4Collaborative Innovation Center of Water Security for Water Source Region of Mid-route Project of South-North Water Diversion of Henan Province, School of Agricultural Engineering, Nanyang Normal University, Nanyang 473061, China

**Keywords:** *Bacillus thuringiensis*, plant growth-promoting bacteria, biochar, heavy metal, radish

## Abstract

The objective of this study was to explore the effect of heavy metal-resistant bacteria and biochar (BC) on reducing heavy metal accumulation in vegetables and the underlying mechanism. We tested *Bacillus thuringiensis* HC-2, BC, and BC+HC-2 for their ability to immobilize Cd and Pb in culture solution. We also studied the effects of these treatments on the dry weight and Cd and Pb uptake of radish in metal-contaminated soils under field conditions and the underlying mechanism. Treatment with HC-2, BC, and BC+HC-2 significantly reduced the water-soluble Cd (34–56%) and Pb (31–54%) concentrations and increased the pH and NH_4_^+^ concentration in solution compared with their vales in a control. These treatments significantly increased the dry weight of radish roots (18.4–22.8%) and leaves (37.8–39.9%) and decreased Cd (28–94%) and Pb (22–63%) content in the radish roots compared with the control. Treatment with HC-2, BC, and BC+HC-2 also significantly increased the pH, organic matter content, NH_4_^+^ content, and NH_4_^+^/NO_3_^−^ ratio of rhizosphere soils, and decreased the DTPA-extractable Cd (37–58%) and Pb (26–42%) contents in rhizosphere soils of radish. Furthermore, BC+HC-2 had higher ability than the other two treatments to protect *radish* against Cd and Pb toxicity and increased *radish* biomass. Therefore, *Bacillus thuringiensis* HC-2 combined with biochar can ensure vegetable safety in situ for the bioremediation of heavy metal-polluted farmland.

## 1. Introduction

Heavy metal pollution is a global phenomenon and has increased with urbanization, industrialization, and modernization of agriculture and animal husbandry [1,2]. Cd and Pb are toxic and biologically non-essential elements in plants and animals [3,4,5]. Cd and Pb in farmland soil not only cause serious damage to soil and plants but also invade the human body through the food chain, thereby endangering human health [6,7,8]. Many agricultural crops, including vegetables, are grown in slightly or moderately heavy metal-contaminated soils. Vegetables may accumulate heavy metals in metal-contaminated soil and thus may pose significantly damage human health and the safety of agricultural products [9,10,11]. As such, searching for an effective way of using and repairing the soil is extremely important.

In situ fixation of heavy metals in soils can perform “production and repair at the same time,” and this fixation is currently applicable to fields with mild heavy metal pollution through bioremediation technology [12,13,14]. The remediation techniques for heavy metal contaminated farmlands mainly include chemical passivation remediation and microbial fixation remediation [15,16]. Chemical passivating agents, such as biochar (BC), can fix heavy metals in soil and reduce the absorption of heavy metals by plants [17,18]. Recently, metal-tolerant and plant growth-promoting bacteria (PGPB) have been reported to reduce available heavy metals in soil and heavy metal uptake in plant tissues. Han et al. [19] reported that the heavy metal-immobilizing bacteria *Serratia liquefaciens* CL-1 and *Bacillus thuringiensis* X30 increased the biomass (25–99%) of and decreased edible tissue Cd and Pb uptake in radish (37–81%). Compared to no inoculation, inoculation with CL-1 and X30 decreased the DTPA-extractable Cd and Pb contents (18–44%) of the rhizosphere soil. PGPB had the ability to secrete indole-3-acetic acid (IAA) and siderophores and improved resistance of plants to heavy metals. They perform fixed precipitation or adsorb heavy metals in the soil and demonstrate “passivation” of their biological effectiveness, thereby reducing metal uptake by plants [20,21,22].

Although the use of microorganisms or BC to prevent vegetables from absorbing heavy metals has been extensively explored, the insight into how microorganisms or BC can reduce the content of available heavy metals in soil remains unclear. In addition, the use of plant biogenic bacteria combined with BC to inhibit the absorption of heavy metals by vegetables in field experiments remains poorly studied. Therefore, to study the mechanisms of heavy metal-immobilizing bacteria inhibiting Cd and Pb uptake in radish and compare the effect of the tested strains and BC on the growth of radish, this study mainly investigated the fixation and adsorption of Cd and Pb by plant growth-promoting bacteria (*Bacillus thuringiensis* HC-2), BC, and *B. thuringiensis* HC-2 combined with BC under solution conditions and their effects on the growth and heavy metal absorption of radish under field conditions. Furthermore, the effects of the strain and BC on the DTPA-extractable Cd and Pb contents, pH, organic matter content, NH_4_^+^-N and NO_3_^−^-N contents and the NH_4_^+^/NO_3_^−^ ratio in the rhizosphere soils of radish were also investigated to evaluate the mechanisms related to such effects.

## 2. Materials and Methods

### 2.1. Test Strain, Biochar and Vegetable

Strain HC-2 was cultured from rhizosphere soils of amaranth using the solid plate method [13]. Strain HC-2 produced IAA (253 mg·L^−1^), 1-aminocyclopropane 1 carboxylate (ACC) deaminase and siderophores, and it resisted Cd (500 mg·L^−1^), Pb (2200 mg·L^−1^), and tetracycline (50 mg·L^−1^). Thus, it could be classified as a heavy metal-resistant bacterium. A 16S rDNA sequence analysis further showed that strain HC-2 belonged to *B. thuringiensis*, with the registration number MH443000. BC was purchased from a transparent fragrant rice processing plant in Yuanyang County. The raw material was a rice husk, which was pyrolyzed at 500 °C for 30 min in a muffle furnace. The BC produced had a particle size of 0.5–1 mm, a pH of 7.84 ± 0.13, and an organic matter content of 29.5 ± 1.2 g·kg^−1^. Meanwhile, Cd and Pb were not detected. A widely-planted Chinese radish variety, the hybridized Qiyehong (*Raphanus raphanistrum subsp.* sativus L., Zhongmou County Seed Company, Zhongmou, China), was used.

### 2.2. Solution Adsorption Test

Cd and Pb fixation and adsorption by strain HC-2 and BC in solution were determined following the method of Chen et al. [13] with appropriate modifications. Triplicate culture flasks containing 50 mL LB liquid medium with 1 mg·L^−1^ Cd and 20 mg·L^−1^ Pb, were prepared and sterilized at 121 °C for 25 min. Strain HC-2 was grown in LB liquid medium at 30 °C, and shaken at 150 rpm for 18 h until the logarithmic phase. Cultures were transferred into sterile centrifuge tubes and centrifuged at 6000 rpm for 10 min to collect the bacteria. The collected bacteria were washed twice using sterile deionized water, resulting in a bacterial suspension with OD_600_ = 1.0. For the HC-2 treatment group, 0.5 mL of HC-2 bacterial suspension was placed in a shaker flask, maintained at 30 °C, and then allowed to stand for 7 days. For the BC treatment group, 0.5 g of BC was added to a shaker flask, maintained at 30 °C, and allowed to stand for 7 days. For the BC+HC-2 treatment group, 0.5 mL of HC-2 bacterial suspension and 0.5 g of BC were added to the shaker flask, maintained at 30 °C, and allowed to stand for 7 days. Meanwhile, a control group (CK) was also prepared, and three replicates were established for each treatment. Samples were collected at days 0, 1, 3, 5, and 7, we removed 7 mL of fermentation broth and centrifuged at 12,000 rpm for 5 min, and the supernatant was analyzed using inductively coupled plasma optical emission spectrometry (ICP-OES, Optima 2100 DV; PerkinElmer, Waltham, MA, USA). The pH of the supernatant was determined by a pH meter (PHS-3CT). Then, the NH_4_^+^ content in the supernatant solution was determined following the method of Jeong et al. [23]. The supernatant (10 mL) was placed in a constant volume tube with 25 mL of deionized water mixed with 0.1 mL of sodium potassium tartrate and 0.5 mL of natron and agitated for 10 min. The absorbance of the solution was measured at 425 nm with water as a reference. The NH_4_^+^ content in the supernatant was measured using a standard curve.

### 2.3. Field Experiment

The field test site was located at the vegetable farm of the Ming Mountain Temple Village (34°49′ N, 113°28′ E) in Zhongmou County Youth Office, Zhengzhou City, Henan Province, P. R. China. The location has an annual average temperature of 17 °C and an annual average precipitation of 1127 mm. The basic physical and chemical properties of the soil were as follows: Uren fine sandy loam, pH 6.91 ± 0.02, 22.4 ± 1.12 g kg^−1^ of organic matter content, 1.58 ± 0.01 g kg^−1^ total N, 1.04 ± 0.01 g kg^−1^ total P, 5.68 ± 0.54 mg kg^−1^ available P, 1.83 ± 0.1 mg kg^−1^ total Cd, and 241 ± 12.3 mg kg^−1^ total Pb. After weeding, turning the ground over, and drying, breaking and leveling the soil, an area of 5 m × 20 m was selected and divided into 12 communities (2 m × 2 m each) using a tape measure. A gap of 50 cm was created between communities. The following four treatment groups were used in the experiment: (1) non-inoculated (CK), (2) inoculated with HC-2, (3) inoculated with BC (BC), and (4) inoculated with the combination of BC and HC-2 strain (BC+HC-2). Each treatment had three replicates. One month before planting, 0.05 kg m^−2^ BC was mixed with the soil in the cultivated layer and dried for 30 days, followed by being evenly distributed on the plot. To reduce the effect of the regional environment on the test results, we randomly determined the corresponding treatment group for each experimental plot. The radish seeds were sterilized, soaked in 5% sodium hypochlorite solution for 5 min, and then washed thrice with sterile deionized water. Some of the radish seeds were placed into a bacterial suspension of strain HC-2 and soaked for 2 h. Then, the seeds were mixed with a biological matrix. The mixture was sown in the field plot for testing. Then, the seedlings were allowed to grow to 2–3 cm, and 12 seedlings were planted in each plot. For inoculation, strain HC-2 was grown in LB medium and agitated at 150 rpm for 18 h until the logarithmic phase. Cultures were transferred into sterile centrifuge tubes and centrifuged at 6000 rpm for 10 min and the collected bacteria were washed twice using sterile deionized water, resulting in a bacterial suspension with OD_600_ = 1.0. The first inoculation was performed when the radish seedlings grew three leaves. The bacterial suspension (150 mL) was inoculated using a 2 cm sterile syringe that injected below the soil surface. The CK group was inoculated with 150 mL of sterile water. After a month of growth, a second inoculation was performed. During irrigation and weeding, turnips were planted on 19 March 2017, and harvested on 20 May 2017.

### 2.4. Sample Processing

Edible parts and leaves of five radishes in each plot were randomly selected, and three groups of samples were treated in each plot. The collected samples were cleaned with sterile deionized water and dried to a constant weight at 80 °C. The Cd and Pb concentrations in the radish were determined following the method of Wang et al. [17]. The dried radish samples were ground with a microplant tissue grinder. Samples (1.00 g) were accurately weighed for microwave digestion using 5% HNO3. The Cd and Pb contents in the digestion solution were determined by ICP-OES. The soil samples were divided into rhizosphere and non-rhizosphere soils. Furthermore, the available Cd and Pb contents in soil were determined following the method of Chen et al. [13] The rhizosphere and non-rhizosphere soil samples (2.0 g each) were separately placed in 50 mL centrifuge tubes, the percentage of soil water was measured, and five extracts received diethylenetriaminepentaacetic acid (DTPA). Then, the samples were alternately extracted and centrifuged at 12,000 rpm for 10 min. The Cd and Pb contents in the supernatant were determined by ICP-AES. Sterile deionized water was added to each soil sample at a soil-to-water ratio of 1:2.5, and the samples were fully oscillated for 2 h and centrifuged at 12,000 rpm for 10 min. The supernatant was collected, and its pH was measured. The organic matter content of potted rhizosphere and non-rhizosphere soil samples under different treatments was determined by the potassium dichromate volumetric method, and the differences were compared. NH_4_^+^-N and NO_3_^−^-N in the rhizosphere soils of pakchoi were measured by spectrophotometric methods with phenol disulfonic acid and indophenol blue reagent, respectively [24].

### 2.5. Colonization Test

The colonization of strain HC-2 in the rhizosphere soil of radish was determined following the methods of He et al. [25]. Then, 0.5 g of rhizosphere soil of radish samples was added to 50 mL of sterile deionized water, and the mixture was placed in a shaker flask containing five glass beads, and agitated at 150 rpm for 30 min. A bacterial suspension (1 mL) was collected, diluted to 10^−4^ and 10^−5^ using LB containing 200 mg L^−1^ Cd, 1500 mg L^−1^ Pb, and 50 mg L^−1^ tetracycline and allowed to develop for 3 days at 30 °C. Fifty colonies from each treatment panel were randomly isolated, purified, and conserved. Genomic DNA was extracted, and the 16S rDNA of each bacterium was amplified by PCR and sent to the Nanjing Jinsirui Sequencing Company (Nanjing, China) for sequencing. According to the species information of 50 strains, the strains *B. thuringiensis* strains can be classified as HC-2, and the proportion of HC-2 in the 50 strains can represent the proportion of HC-2 in the rhizosphere soil of each treatment group.

### 2.6. Data Processing

One-way analysis of variance and Tukey’s test (*p* < 0.05) were used to compare the averages of dry weight, Cd and Pb contents in the radish and soils, soil pH, organic matter, NH_4_^+^-N content and NO_3_^−^-N content in the presence of strain HC-2 and BC with those of the controls. The impact of pH on the Cd and Pb concentrations in the solution was also compared. Statistical analyses were performed using SAS 8.2 (Statistical Analysis System, Cary, NC, USA)

## 3. Results

### 3.1. Effects of the Strain HC-2 and BC on Cd and Pb Immobilization, pH and NH_4_^+^ Concentration in Solution

The adsorption and immobilization of Cd and Pb by the tested strain and BC were investigated by a static culture method. As shown in Figure 1a,b, the concentrations of Cd (0.99–0.97 mg L^−1^) and Pb (19.87–19.82 mg L^−1^) in the control group showed no significant change as the culture time increased. However, HC-2, BC, and BC+HC-2 treatments significantly (*p* < 0.05) decreased the concentrations of Cd (34–56%) and Pb (31–54%) in the solution. On day 1 of culture, the ability of BC to fix and adsorb Cd and Pb was obviously higher than that of strain HC-2. However, on day 5, the ability of the HC-2 strain to immobilize Cd and Pb was obviously higher than that of BC. In addition, compared with the BC treatment, the BC+HC-2 treatment significantly reduced the concentrations of Cd and Pb by 17–33% and 22–32%, respectively, indicating that the BC+HC-2 treatment group had a stronger ability to fix and adsorb Cd and Pb in solution than the other groups. As shown in Figure 1c, compared with the control, the HC-2, BC, and BC+HC-2 treatment groups significantly (*p* < 0.05) increased the pH of the solution. The pH values of solutions treated with HC-2 and BC+HC-2 increased gradually. On day 7, the maximum pH values were reached in the HC-2 (7.89) and BC+HC-2 (8.13) treatment groups. The pH of the solution in the BC treatment group reached its highest level (7.42) on day 3 and remained stable. As shown in Figure 1d, compared with that of the control, the NH_4_^+^ concentration in the solutions treated with either HC-2 or HC-2+BC increased gradually. On day 7, the HC-2 and BC+HC-2 treatment groups had NH_4_^+^ concentrations of 29.34 and 36.21 mg L^−1^, respectively. Meanwhile, the NH_4_^+^ content in the solution of the BC treatment group peaked on day 1 and remained stable thereafter. Thus, the HC-2 strain can produce the alkaline substance NH_4_^+^ in the culture process, thereby gradually increasing the pH of the solution. Furthermore, BC+HC-2 treatment had a stronger capacity to produce NH_4_^+^ than treatment with strain HC-2 alone.

### 3.2. Effects of Different Treatments on the Biomass and Heavy Metal Content of Radish

As observed in Figure 2, compared with the control treatment, strain HC-2, BC, and BC+HC-2 significantly (*p* < 0.05) increased the dry weight of radish roots (25.2–54.6%) and leaves (104–186%). Compared with HC-2 or BC, BC+HC-2 significantly improved the dry weight of radish roots (18.4–22.8%) and leaves (37.8–39.9%), indicating that the combined application of strain HC-2 and BC could better improve the biomass of radish. Compared with the control treatment, strain HC-2, BC, and BC+HC-2 significantly (*p* < 0.05) reduced the Cd (43.6–82.9%) and Pb (37.5–63.2%) contents in radish tissues. The contents of Cd and Pb in radish roots and leaves from the BC+HC-2 treatment group were significantly (*p* < 0.05) lower than those from the HC-2 and BC treatment groups. Thus, the combined application of strain HC-2 and BC can reduce the Cd and Pb contents in radish more than application of strain HC-2 or BC alone.

### 3.3. Effects of Different Treatments on the DTPA-Extractable Cd and Pb Contents in the Rhizosphere and Bulk Soils of Radish

The tested strain HC-2 and BC affect the DTPA-extractable Cd and Pb contents in the rhizosphere and bulk soils of radish, and the specific results are shown in Figure 3. According to Figure 3, compared with the control, strain HC-2, BC, and BC+HC-2 significantly reduced the DTPA-extractable Cd (37–58%) and Pb (26–42%) contents in the rhizosphere soil of radish. At the same time, the DTPA-extractable Cd (10–12%) and Pb (10–11%) contents in the bulk soil were also improved with treatment with HC-2, BC, and BC+HC-2, but, the DTPA-extractable Cd and Pb contents in the rhizosphere soil were significantly lower than those in the bulk soil. The DTPA-extractable Cd and Pb contents in the rhizosphere soil of radish in the BC+HC-2 treatment group were significantly lower (*p* < 0.05) than those in the HC-2 and BC treatment groups.

### 3.4. Effects of Different Treatments on pH and Organic Matter Content in the Rhizosphere and Bulk Soils of Radish

The contents of DTPA-extractable Cd and Pb in the soil are closely related to the soil pH and organic matter content. In this study, the pH and organic matter content in the radish rhizosphere and bulk soils under different treatments were determined, and the results are shown in Figure 4. Compared with the control, strain HC-2, BC, and BC+HC-2 significantly (*p* < 0.05). increased the pH (6.87 to 7.05–7.15) and organic matter content (27–45%) in the rhizosphere soil of radish. Strain HC-2 had no significant effect on the pH and organic matter content of the bulk soil. However, BC and BC+HC-2 significantly (*p* < 0.05) increased the pH and organic matter content in the bulk soil of radish, and the pH and organic matter content of the rhizosphere soil were significantly higher than those of the bulk soil.

### 3.5. Change in the NH_4_^+^-N and NO_3_^−^-N Contents and the NH_4_^+^/NO_3_^−^ Ratio in the Rhizosphere Soils of Radish

The NH_4_^+^-N and NO_3_^−^-N contents in soil are key factors affecting the soil pH. Therefore, the effects of different treatments on the NH_4_^+^-N and NO_3_^−^-N contents and the NH_4_^+^/NO_3_^−^ ratio in the rhizosphere soils of the radish are shown in Figure 5. Compared with the control, treatment with HC-2, BC, and BC+HC-2 significantly (*p* < 0.05) increased the NH_4_^+^-N (69.1–141%) and NO_3_^−^-N (22.3–41.3%) contents in the rhizosphere soils of the radish. Furthermore, the NH_4_^+^-N and NO_3_^−^-N contents in the rhizosphere soils treated with BC+HC-2 were higher than those in rhizosphere soils receiving other treatments. The NH_4_^+^/NO_3_^−^ ratio was 1.64 in the control treatment soil; however, the NH_4_^+^/NO_3_^−^ ratios were 3.15, 2.04, and 3.1 in the soils treated with HC-2, BC, and BC+HC-2, respectively. These data revealed that the ability of HC-2 and BC to increase the NH_4_^+^-N content was higher than their ability to increase the NO_3_^−^-N content in the rhizosphere soils.

### 3.6. Colonization of the Tested Strains in the Rhizosphere Soil of Radish

Through 16S rRNA gene sequencing and biochemical characterization, we found a large number of strains in the rhizosphere soil receiving treatments HC-2 and BC+HC-2, as shown in Table 1, which shows that the indicated strains could colonize the rhizosphere soil of radish. Notably, in the rhizosphere soils, the proportion of HC-2 in the rhizosphere soils treated with HC-2 (31.6%) and BC+HC-2 (43.3%) was higher than the proportion of HC-2 in the control (9.6%) and BC (11.2%) rhizosphere soils, suggesting that strain HC-2 could colonize the rhizosphere soil of radish well.

## 4. Discussion

In our research, the heavy metal-resistant bacterium *Bacillus thuringiensis* HC-2, biochar and a combination of the strain HC-2 with BC were studied for their ability to immobilize Cd and Pb in culture solution. Treatment with HC-2, BC, and BC+HC-2 significantly reduced water-soluble Cd and Pb concentrations and increased the pH and NH_4_^+^ concentration in solution compared with their control values. The results also showed that strains HC-2 and BC immobilized Cd and Pb by extracellular adsorption and increasing the pH and NH_4_^+^ concentration of the solution (Figure 1). HC-2, BC, and BC+HC-2 treatments improved the biomass and reduced Cd and Pb accumulation of the radish (Figure 2). These treatments also significantly decreased the DTPA-extractable Cd and Pb contents and increased the pH, organic matter content, NH_4_^+^-N and NO_3_^−^-N contents, and NH_4_^+^/NO_3_^−^ ratio of the rhizosphere soils (Figure 3, Figure 4 and Figure 5). Strain HC-2 combined with BC had a higher ability to reduce the radish Cd and Pb uptake and the DTPA-extractable Cd and Pb contents in the rhizosphere soils of pakchoi than strain HC-2 alone or BC.

Bacteria reduce the bioavailability and the concentrations of heavy metal ions in solution by adsorbing them [26]. Bacteria can eliminate heavy metals through adsorption in their cell walls, complexation with phosphates, and precipitation or adsorption of heavy metals by other anions produced by bacterial metabolism [27]. Heavy metal ions interact with chemical groups on microbial surface proteins, polysaccharides and lipids to form metal complexes, which are adsorbed and fixed on cell surfaces. Extracellular polymeric substances (EPS), mainly composed of protein and polysaccharides, are the main components of sludge organics. EPS traditionally exhibit a porous polymeric structure, and polar heavy metals can be easily adsorbed onto adsorption sites (by their functional groups such as carboxyl and hydroxyl groups) on the surface of EPS. Overall, the physicochemical structural characteristics of EPS significantly affect the adsorption processes of heavy metals. Sheng et al. [28] demonstrated that the oxygen atoms in the carboxyl groups of proteins contributed to the adsorption removal of Cu^2+^. A previous study by Wei et al. [29] found that the complexation of Zn(II) with O–H and N–H groups and C-O stretching significantly decreased the fluorophore intensity of protein- and humic-like substances. In our study, strain HC-2 significantly (*p* < 0.05) decreased the concentrations of Cd and Pb in solution. BC is an alkaline, good adsorption material with a large porosity and specific surface area, and it can strongly adsorb and immobilize heavy metals [30]. In our study, with increasing culture time, the Cd and Pb concentrations in solution were significantly reduced by HC-2, BC, and BC+HC-2. Beatriz et al. [31] discovered that *Lactococcus lactis* can generate putrescine, which increases the pH of the medium, turning it alkaline. In our study, HC-2 and BC+HC-2 improved the pH of the fermentation broth. On day 7 of culture, the pH in the solution of the strain HC-2-treated samples was 7.89, whereas that of the BC+HC-2 samples was 8.13. During the growth process, strain HC-2 produced some alkaline substances, such as NH_4_^+^, thereby increasing the pH of the solution to cope with the stress of heavy metals.

The chemical passivating agent biochar is a good adsorption material. Adding it in farmland soil as a passivating agent to improve soil quality and absorb pollutants not only reduces the effectiveness of heavy metals but also reduces soil nutrient leaching, improves soil quality, and increases crop yield [32,33,34]. Chen et al. [35] reported that the effect of biochar on heavy metal concentrations in plants varied depending on soil properties, biochar type, plant species, and metal contaminants. In a survey [35], a total of 1298 independent observations were collected from 74 published papers, and biochar addition to soils was shown to result in average decreases of 38%, 39%, 25% and 17%, respectively, in the accumulation of Cd, Pb, Cu and Zn in plant tissues. In our study, BC significantly reduced the effective Cd and Pb contents in the radish soil, reduced the Cd and Pb contents in radish, and improved the biomass of radish. The application of chemical passivators such as biochar can immobilize heavy metals in soil and reduce the uptake of heavy metals by plants. However, long-term use of chemical passivators may lead to changes in soil physical and chemical properties, decreases in the number and activity of soil microorganisms and degradation of soil fertility [36]. Under the stress of heavy metals, some bacteria enter and cooperate with host plants to resist the stress of heavy metals. Endophytes have many excellent characteristics, such as the absorption and tolerance of heavy metals. The advantages of endophytic bacteria have attracted the attention of researchers [37,38]. Studies have shown that plant growth-promoting bacteria can reduce the toxicity of heavy metals to plants, secrete substances that promote plant growth, and change the uptake and transport of heavy metals by plants [13,37,38]. Therefore, plant growth is promoted after plants have been inoculated with plant growth-promoting bacteria. In addition, the efficiency of heavy metal damage repair by plants themselves has been improved. Therefore, compared with chemical regulation, PGPB have greater potential and advantages in regulating plant growth because PGPB regulation releases no pollution into the environment and is low risk, very cheap, and more closely related to plants. In addition, phytogenic bacteria not only promote plant growth but also immobilize heavy metals in soil and reduce the effective state of heavy metals [38,39]. Given the complexity of the natural environment, microbial passivation and fixation of soil heavy metals has a huge effect. Considering the influence of environmental conditions, functional strains cannot easily survive, whereas BC causes the soil to harden easily and is conducive to plant growth. The association of microorganisms with BC can compensate for the shortcomings of each. In addition, it can increase crop yield, reduce the heavy metal content of agricultural products, and improve the agricultural ecological environment of the soil. Therefore, combined microbial-chemical modifier repair technology has become a hot research topic. In this study, both HC-2 and BC+HC-2 treatments significantly reduced the Cd and Pb contents in radish tissues. Wang et al. [40] reported that *Bacillus subtilis* 38 combined with organic fertilizer could significantly reduce Cd and Zn concentrations in the edible parts of lettuce, radish, and soybean. The combined application (BC+HC-2) had better effects on improving the edible radish biomass and reducing the Cd and Pb contents in the edible part of radish than the sole application of BC or functional strain HC-2.

Soil pH reflects the levels of acid and alkaline compounds in the soil, and it is the most critical factor affecting the distribution, transformation, and bioavailability of heavy metals in soil [41]. It can also affect the adsorption, complexation, and hydrolysis equilibrium of heavy metals in the soil, thereby affecting heavy metal absorption in plants [42]. Chen et al. [13] reported that *Neorhizobium huautlense* T1-17 significantly increased the pH and reduced the available contents of Cd and Pb in a heavy metal-contaminated rhizobium soil. In the current study, each treatment significantly increased the pH and organic matter content of the radish rhizosphere soil, thereby reducing the content of available heavy metals. Nitrogen (N) management is a promising agronomic strategy to minimize Cd and Pb contamination in crops [43,44]. Some scholars report that when NO_3_^−^ is taken up by plants, there is a simultaneous uptake of protons (H^+^), resulting in an increase in rhizosphere pH. When NH_4_^+^ is taken up, H^+^ is released into the rhizosphere, resulting in a decrease in rhizosphere pH, suggesting that compared with NO_3_^−^ fertilizers, NH_4_^+^ fertilizers could result in enhanced Cd uptake due to a decrease in soil pH [43,45]. However, contrary evidence has been obtained in several other studies. Xie et al. [46] found that *Thlaspi caerulescens* plants fed NO_3_^−^ accumulated much more Cd than the corresponding plants supplied with NH4+. Mao et al. [44] showed that cadmium inhibits nitrate transporter 1.1 (NRT1.1)-mediated nitrate (NO3-) uptake in Arabidopsis (*Arabidopsis thaliana*) and impairs NO_3_^−^ homeostasis in roots. In our study, treatment with HC-2, BC, and BC+HC-2 significantly increased the NH_4_^+^-N content and the NH_4_^+^/NO_3_^−^ ratio in the rhizosphere soils, resulting in decreased DTPA-extractable Cd and Pb contents in the rhizosphere soil and Cd and Pb uptake by radish.

In this study, we focused on the effect of the treatment with BC+HC-2 on the accumulation of Cd and Pb in radish (edible tissue) in Cd- and Pb-polluted soils. The results showed that the edible tissue Cd (0.06 ± 0.01 mg kg^−1^ of fresh weight) and Pb (0.07 ± 0.02 mg kg^−1^ of fresh weight) contents of the radish were lower than the Cd and Pb limit standards of fresh weight of radish (0.1 mg kg^−1^) set by the European Commission. (2011) [47] in the presence of BC+HC-2. The results suggest that strain HC-2 combined with BC can guarantee the safety levels of heavy metals in vegetables in heavy metal-contaminated farmland.

## 5. Conclusions

The results show that the heavy metal-resistant bacterium *Bacillus thuringiensis* HC-2, biochar, and biochar combined with *Bacillus thuringiensis* HC-2 reduced the concentrations of Cd and Pb by extracellular adsorption and increasing the pH in the solution. In the growth process, they also produced some alkaline substances, such as NH_4_^+^, increased the pH of the fermented liquid, and reduced heavy metal toxicity. In the field experiment, HC-2, BC, and BC+HC-2 treatments significantly promoted the growth of radish and inhibited the absorption of Cd and Pb in radish tissues. HC-2, BC, and BC+HC-2 treatments alleviated Cd and Pb toxicity, promoted growth, and reduced Cd and Pb uptake of radish by reducing the DTPA-extractable Cd and Pb contents and increasing the pH, organic matter content, NH_4_^+^-N content, and NH_4_^+^/NO_3_^−^ ratio in the rhizosphere soils of radish. Furthermore, in the presence of BC+HC-2, the Cd and Pb contents in the radish roots reached the food safety standard set by the European Commission. Our results suggest that biochar combined with *Bacillus thuringiensis* HC-2 can play an important role in improving the yield and reducing the heavy metal content in radish or other vegetables.

## Figures and Tables

**Figure 1 ijerph-16-03676-f001:**
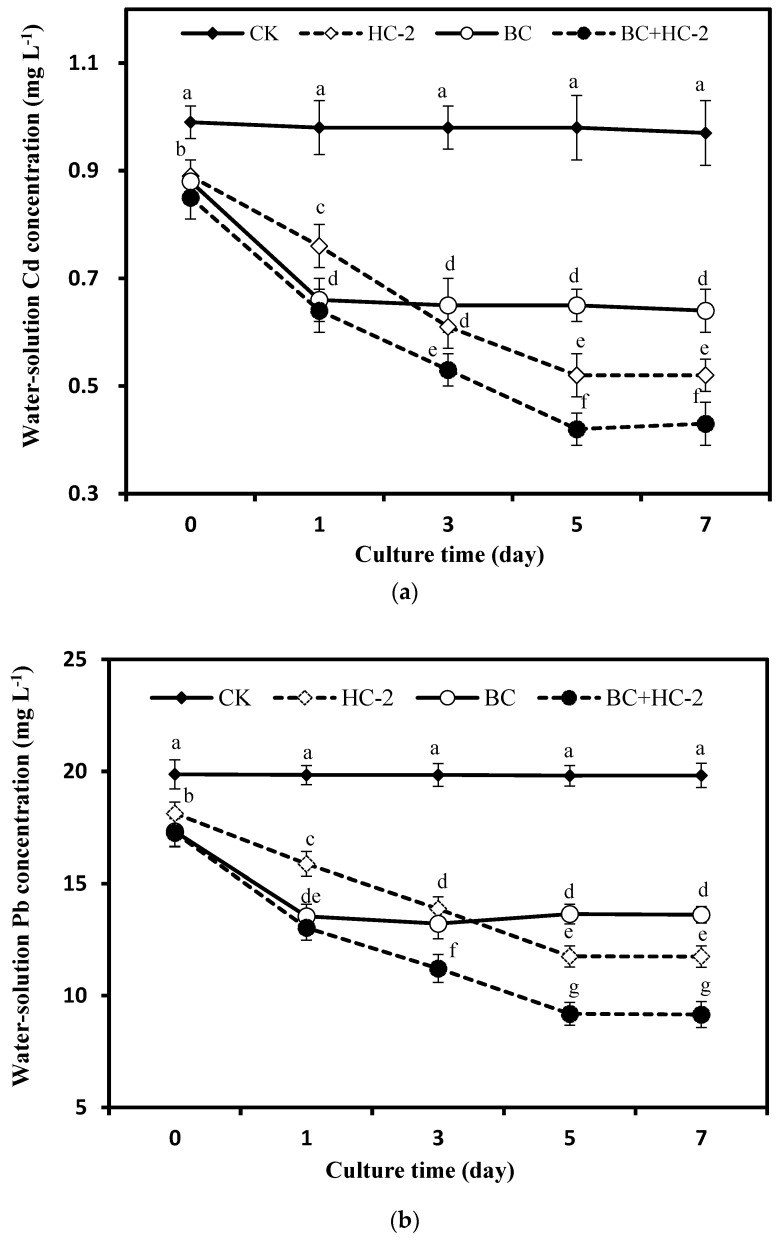
Changes in Cd and Pb concentrations, pH, and NH_4_^+^ concentrations in the culture solution with strain HC-2 and BC. (**a**): the influence of different treatments on the concentration of water-solution Cd. (**b**): the influence of different treatments on the concentration of water-solution Pb. (**c**): the influence of different treatments on the pH of the solution. (**d**): the influence of different treatments on the NH_4_^+^ concentration of the solution. Error bars are ± standard error (*n* = 3). Data followed by the different letters (a–h) are significantly different (*p* < 0.05) according to Tukey’s test.

**Figure 2 ijerph-16-03676-f002:**
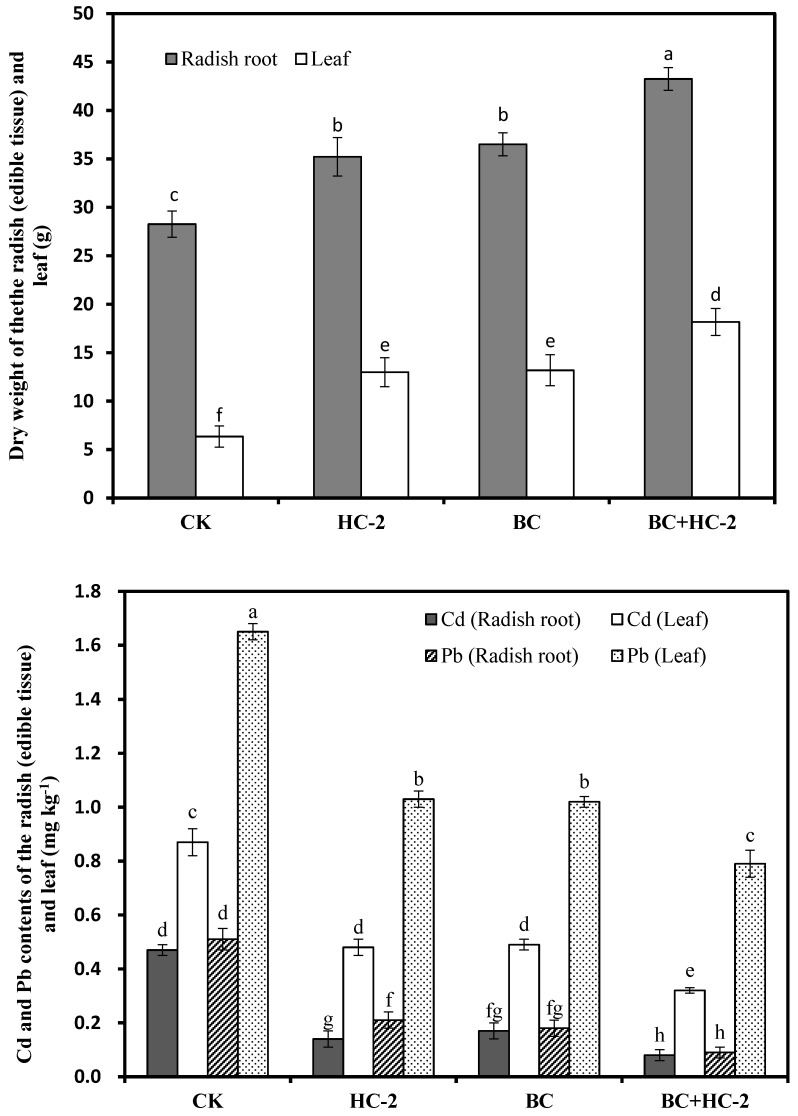
Effects of strain HC-2, BC, and BC+HC-2 on the dry weight and heavy metal accumulation of the radish roots and leaf. Error bars are ± standard error (*n* = 3). Data followed by the different letters (a–h) are significantly different (*p* < 0.05) according to Tukey’s test.

**Figure 3 ijerph-16-03676-f003:**
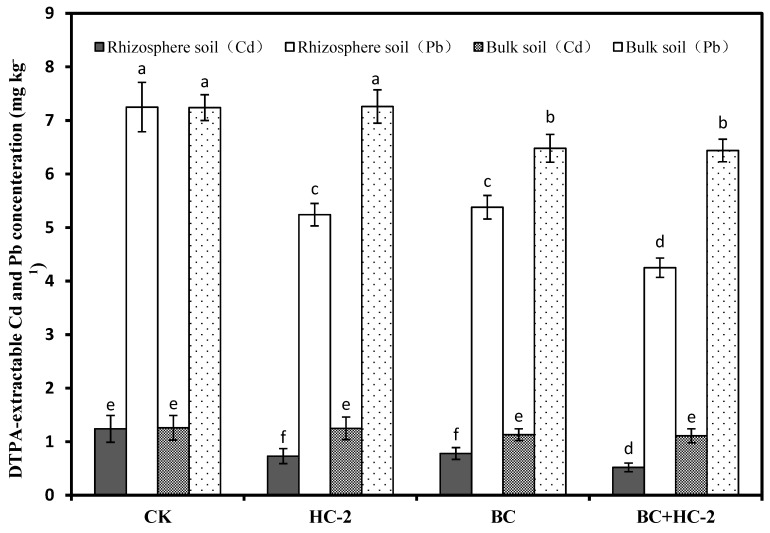
Effects of strain HC-2, BC, and BC+HC-2 on the DTPA-extractable Cd and Pb contents in the rhizosphere and bulk soils of radish. Error bars are ± standard error (*n* = 3). Bars indicated by the different letters for each Cd and Pb contents were significantly (*p* < 0.05) different according to Tukey’s test.

**Figure 4 ijerph-16-03676-f004:**
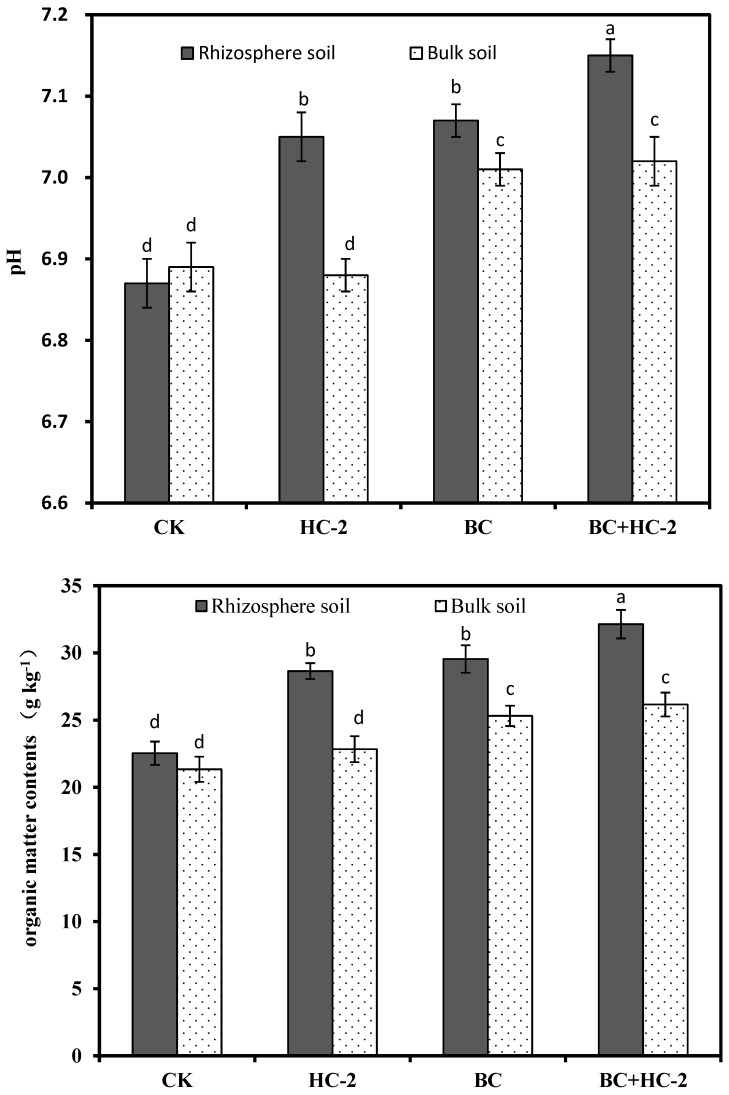
Effects of strain HC-2, BC, and BC+HC-2 on the pH and organic matter contents in the rhizosphere and bulk soils of radish. Error bars are ± standard error (*n* = 3). Bars indicated by the different letter within each pH and organic matter contents were significantly (*p* < 0.05) different according to Tukey’s test.

**Figure 5 ijerph-16-03676-f005:**
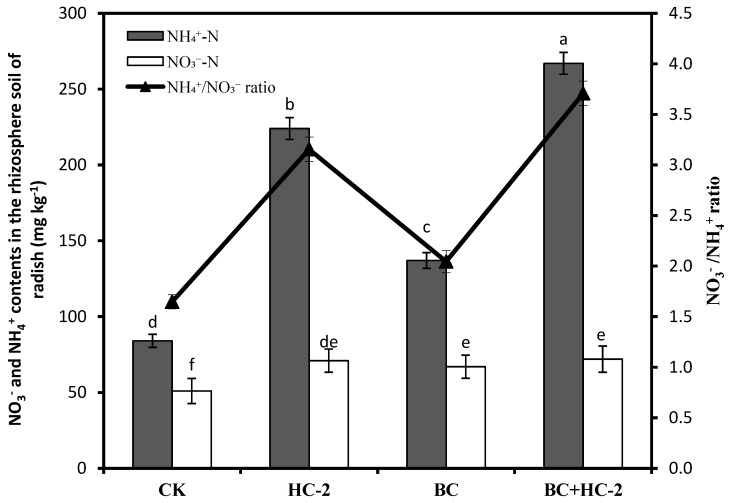
Effects of treatments of strain HC-2, BC, and BC+HC-2 on the NH_4_^+^-N content, NO_3_^−^-N content, and the NH_4_^+^/NO_3_^−^ ratio in the rhizosphere soils of radish. Error bars are ± standard error (*n* = 3). Bars indicated by the different letter within each NH_4_^+^-N and NO_3_^−^-N contents were significantly (*p* < 0.05) different according to Tukey’s test.

**Table 1 ijerph-16-03676-t001:** The proportion of the tested strain in the rhizosphere soil of radish.

	CK	HC-2	BC	BC+HC-2
The proportion of strain HC-2 *	9.6% ± 0.8% ^c^	31.6% ± 1.8% ^b^	11.2% ± 1.1% ^c^	43.3% ± 2.2% ^a^

* Data followed by the different letters (a–c) within the same line was significantly different (*p* < 0.05) according to Tukey’s test. Average (±, standard deviations) of the proportion of strain HC-2 from three repetitive soil samples.

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
