# Peer review of "Effects of Bacillus thuringiensis HC-2 Combined with Biochar on the Growth and Cd and Pb Accumulation of Radish in a Heavy Metal-Contaminated Farmland under Field Conditions"

_ijerph, 2019, doi:10.3390/ijerph16193676_

Round 1

Reviewer 1 Report

The study is well executed and important. Heavy metals are a problem for agriculture in many parts of the world, and are a well-documented problem for the PRC. I have no problem with the science in the study. It builds upon previous studies using bacteria or biochar for remediation, although this is the only one I know that combines both. 

I made a number of comments to improve the writing. I did so using "track changes in Word, converted that file to a pdf, and have uploaded it here for the authors to use. English is a difficult language, and I hope they persist in trying to publish in my native tongue. There are services out there that will help with writing, and they could consider finding collaborators that might be of assistance. But the quality of their science warrants publication. 

Author Response

Reviewer #1: The study is well executed and important. Heavy metals are a problem for agriculture in many parts of the world, and are a well-documented problem for the PRC. I have no problem with the science in the study. It builds upon previous studies using bacteria or biochar for remediation, although this is the only one I know that combines both.

I made a number of comments to improve the writing. I did so using "track changes in Word, converted that file to a pdf, and have uploaded it here for the authors to use. English is a difficult language, and I hope they persist in trying to publish in my native tongue. There are services out there that will help with writing, and they could consider finding collaborators that might be of assistance. But the quality of their science warrants publication

Response: Thank you for your suggestion, I had revised all the places you marked and the revised manuscript had been edited by native English speaker at American Journal Experts.

Reviewer 2 Report

General

Remediating the heavy metal pollution of soils using some environmental-friendly microbiological strains is very promising techniques. The authors found that HC-2 is very effective to fix dissolved heavy metals and to reduce the accumulation of heavy metals in the crops, and then designed a plot experiment with three treatments, HC-2, BC and BC+HC-2. The results indicated that all the three treatments reduced the contents of Cd and Pb in radish and increased its biomass. Moreover, the combined treatment of BC+HC-2 seemed more effective for remediation. BC is biochar, which, of course, can adsorb heavy metals in solution and reduce their toxicity to crops. HC-2 belongs to B. thuringiensis. It is interesting that the strains can reduce and ease the toxicity of heavy metals to crops.

In general, the manuscript is scientifically meaningful and the English expression is basically fluent. Its content is also suitable to the publication in the IJERPH. I suggest acceptance with minor revision.

Special

Line89: mg L-1. Use superscript.

Line 176: bystatic?

Line 189,192,193, 195: NH4+. Use superscript and subscript.

Line 191: m L-1. Use superscript.

Line 254: NH4+ and NO3-?

Line 263: N3?

-----

Author Response

Reviewer #2: General, remediating the heavy metal pollution of soils using some environmental-friendly microbiological strains is very promising techniques. The authors found that HC-2 is very effective to fix dissolved heavy metals and to reduce the accumulation of heavy metals in the crops, and then designed a plot experiment with three treatments, HC-2, BC and BC+HC-2. The results indicated that all the three treatments reduced the contents of Cd and Pb in radish and increased its biomass. Moreover, the combined treatment of BC+HC-2 seemed more effective for remediation. BC is biochar, which, of course, can adsorb heavy metals in solution and reduce their toxicity to crops. HC-2 belongs to B. thuringiensis. It is interesting that the strains can reduce and ease the toxicity of heavy metals to crops.

In general, the manuscript is scientifically meaningful and the English expression is basically fluent. Its content is also suitable to the publication in the IJERPH. I suggest acceptance with minor revision.

Line89: mg L-1. Use superscript.

Response: Thank you for your suggestion, I had revised all the superscript problems.

bystatic?

Response: I'm very sorry, it was “by static”. It has been revised in the text.

Line 189,192,193, 195: NH4+. Use superscript and subscript.

Response: Thank you for these reference format editing suggestions, these problems have been corrected.

Line 191: m L-1. Use superscript.

Response: This problem has been corrected.

Line 254: NH4+ and NO3-?

Response: I'm very sorry. This should be “NH4+ -N and NO3- -N”.

Line 263: N3?

Response: I'm very sorry. This should be “HC-2”

Reviewer 3 Report

Comments on Manuscript ID: ijerph-573039: Effects of Bacillus thuringiensis HC-2 combined with biochar on the growth and Cd and Pb accumulation of radish in the heavy metal-contaminated  farmland under field condition,  submitted by Zigang Li, Peng Wang, Xiaoyu Yue, Jingtao Wang, Baozeng Ren, Lingbo Qu, Hui Han

The manuscript needs substantial improvement. Comments are listed below:

The introductory part is too narrative and it is hard to understand the necessity of the present study. More concise and informative verbalisation is needed.

Line 41: “Vegetables easily absorb heavy metals.” – The sentence need to be developed  further.

Line 52-56: This part is difficult to comprehend. Please reformulate it.

Line 66-72: This is a brief description of the experimental approach. A clear research question and hypotheses are missing.

Line 154,  256 – 255: Incorrectly written record form “NH4+-N and NO3--N” 

The statistical analysis should be verified. The significant effects presented in some (Fig. 2, 3, 4 and 5) charts do not seem to be justified. Please explain them.

There is no information concerning the OD of the bacteria applied to soil.

There is no reference samples of analyzed Cd and Pb for plants and soils.

Line 312-315: This part is difficult to comprehend.

Line 298, 326, 346: Lactococcus lactis, Bacillus subtilis ect – use italics form.

The academic stile should be used throughout the manuscript, so forms like "we" should be avoided. English should also be carefully checked and corrected, preferably

Author Response

Reviewer #3: Comments on Manuscript ID: ijerph-573039: Effects of Bacillus thuringiensis HC-2 combined with biochar on the growth and Cd and Pb accumulation of radish in the heavy metal-contaminated farmland under field condition, submitted by Zigang Li, Peng Wang, Xiaoyu Yue, Jingtao Wang, Baozeng Ren, Lingbo Qu, Hui Han

The manuscript needs substantial improvement. Comments are listed below:

1.The introductory part is too narrative and it is hard to understand the necessity of the present study. More concise and informative verbalisation is needed.

Response: Thank you for your suggestion. I have carefully revised the introductory to make it concise and clear.

Line 41: “Vegetables easily absorb heavy metals.” – The sentence need to be developed further.

Response: Thank you for your suggestion. I had developed this sentence.

Many agricultural crops, including vegetables, are grown in slightly or moderately heavy metal-contaminated soils. Vegetables may accumulate heavy metals in metal-contaminated soil and thus may pose significantly damage human health and the safety of agricultural products.

Line 52-56: This part is difficult to comprehend. Please reformulate it.

Response: Thank you for this helpful suggestion, I had reformulated this part. PGPB had the ability to secrete indole-3-acetic acid (IAA) and siderophores and improved resistance of plants to heavy metals. They perform fixed precipitation or adsorb heavy metals in the soil and demonstrate "passivation" of their biological effectiveness, thereby reducing metal uptake by plants”.

Line 66-72: This is a brief description of the experimental approach. A clear research question and hypotheses are missing.

Response: Thank you for your suggestion. I had already added research question and hypotheses.

Although the use of microorganisms or BC to prevent vegetables from absorbing heavy metals has been extensively explored, the insight into how microorganisms or BC can reduce the content of available heavy metals in soil remains unclear. In addition, the use of plant biogenic bacteria combined with BC to inhibit the absorption of heavy metals by vegetables in field experiments remains poorly studied. Therefore, to study the mechanisms of heavy metal-immobilizing bacteria inhibiting Cd and Pb uptake in radish and compare the effect of the tested strains and BC on the growth of radish, this study mainly investigated the fixation and adsorption of Cd and Pb by plant growth-promoting bacteria (Bacillus thuringiensis HC-2), BC, and B. thuringiensis HC-2 combined with BC under solution conditions and their effects on the growth and heavy metal absorption of radish under field conditions. Furthermore, the effects of the strain and BC on the DTPA-extractable Cd and Pb contents, pH, organic matter content, NH4+-N and NO3--N contents and the NH4+/NO3- ratio in the rhizosphere soils of radish were also investigated to evaluate the mechanisms related to such effects.

Line 154, 256 – 255: Incorrectly written record form “NH4+-N and NO3--N”

Response: This problem has been corrected.

The statistical analysis should be verified. The significant effects presented in some (Fig. 2, 3, 4 and 5) charts do not seem to be justified. Please explain them.

Response: One-way analysis of variance and Tukey's test (P < 0.05) were used to compare the averages of dry weight, Cd and Pb contents in the radish and soils, soil pH, organic matter , NH4+- N content and NO3--N content in the presence of strain HC-2 and BC with those of the controls. The impact of pH on the Cd and Pb concentrations in the solution was also compared. Statistical analyses were performed using SAS 8.2 (Statistical Analysis System, USA)

There is no information concerning the OD of the bacteria applied to soil.

Response: For inoculation, strain HC-2 was grown in LB medium and agitated at 150 rpm for 18 h until the logarithmic phase. Cultures were transferred into sterile centrifuge tubes and centrifuged at 6,000 rpm for 10 min and the collected bacteria were washed twice using sterile deionized water, resulting in a bacterial suspension with OD600 = 1.0

There is no reference samples of analyzed Cd and Pb for plants and soils.

Response: In all the analyses, the non-inoculated treatment group was used as the control group. The Cd and Pb concentrations in the plants were analyzed according to the method of Wang et al. (2016). Briefly, roots and leaves were separated and washed to remove any nonspecifically bound Cd and Pb. Roots and leaves were oven-dried. The oven-dried samples were ground using a stainless steel mill for analysis. Subsamples of root and leaf samples were then digested in a mixture of concentrated HNO3 and HClO4 (4:1, v/v). Cd and Pb concentrations in the samples were analyzed using ICP-OES. Reagent blank and analytical duplicates were used where appropriate to ensure accuracy and precision in the analysis. Rhizosphere and bulk soils of the radishes were collected The DTPA (0.05 M)-extractable Cd and Pb contents in the rhizosphere and bulk soils were determined by ICP-OES (Chen et al., 2016).

Line 312-315: This part is difficult to comprehend.

Response: I mean, the application of chemical passivators such as biochar can immobilize heavy metals in soil and reduce the uptake of heavy metals by plants. However, long-term use of chemical passivators may lead to changes in soil physical and chemical properties, decrease in the number and activity of soil microorganisms and degradation of soil fertility. Biochar alone is not the best choice. Combining strains with biochar can solve these shortcomings.

Line 298, 326, 346: Lactococcus lactis, Bacillus subtilis ect – use italics form.

Response: This problem has been corrected.

The academic stile should be used throughout the manuscript, so forms like "we" should be avoided. English should also be carefully checked and corrected, preferably

Response: Thank you for language editing suggestions, the revised manuscript has been edited by native English speaker at American Journal Experts.

Reviewer 4 Report

Manuscript ID: ijerph-573039 

Type of manuscript: Article
Title: Effects of Bacillus thuringiensis HC-2 combined with biochar on the
growth and Cd and Pb accumulation of radish in the heavy metal-contaminated
farmland under field condition

The present study focused on exploring the effect and mechanisms of heavy metal-resistant bacteria and biochar (BC) in reducing heavy metal accumulation in vegetables, and according to the conclusions Bacillus thuringiensis HC-2”, biochar and biochar combined with B. thuringiensis reduced the cd and Pb concentrations

 As a reviewer I have mayor concern about the present article

Why the authors never discuss along the introduction or in discuss section a previous research work entitled Metal-immobilizing Serratia liquefaciens CL-1 and Bacillus thuringiensis X30 increase biomass and reduce heavy metal accumulation of radish under field conditions, this work that was previous published in 2018 reach similar conclusions the only variant is the thuringiensis strain and the use of other bacteria termed Serratia liquefaciens. Therefore, the originality of the present study with respect to the other in terms of the findings it refers, and their usefulness is not clear, since when reading both it seems that the conclusions are the same.

It is not understood why to include in the work development a group in Figure 2 where the existence of the thuringiensis strain designated as x-30 is mentioned, which was evaluated in the previous work published in 2018. It is important to mention that This strain was never described or mentioned in the M and M section where only strain HC-2 is included. So, it is not clear, why a result was included in the figures of a strain that was not evaluated?

Specific comments

1.- In Figure 1 (page 6.7) there is no statistical analysis, so it is not possible to establish differences or be able to draw a conclusion of the results.

2.-On page 9 L232 the P value is P<0.05 on the other hand in the foot note of Figure 3 the p value mentioned is   P>0.05 pleas homogenize this change can led to confusion review in all figures.

3.-In Table 1 lack of statistical analysis was observer, moreover, the strain HC-2 is in all groups including the control. How is possible to found differences between groups?

4.-Please discuss the role of strain HC2 as endophytic bacteria.

5.- Is important to discuss in more detail the relationship between the B. thuringienis proteins and the heavy metal absorption.

6.- Please carefully review the spelling and English along the document in order to avoid mistakes i.e L 275 pag 1” in the group of contral” should be “In the control group”

7.- I strongly recommended the article need to be reviewed by a statistical expert and a English native speaker.

Author Response

Reviewer #4: Manuscript ID: ijerph-573039

Title: Effects of Bacillus thuringiensis HC-2 combined with biochar on the
growth and Cd and Pb accumulation of radish in the heavy metal-contaminated
farmland under field condition

The present study focused on exploring the effect and mechanisms of heavy metal-resistant bacteria and biochar (BC) in reducing heavy metal accumulation in vegetables, and according to the conclusions Bacillus thuringiensis HC-2”, biochar and biochar combined with B.thuringiensis reduced the Cd and Pb concentrations.

As a reviewer I have mayor concern about the present article

Why the authors never discuss along the introduction or in discuss section a previous research work entitled “Metal-immobilizing Serratia liquefaciens CL-1 and Bacillus thuringiensis X30 increase biomass and reduce heavy metal accumulation of radish under field conditions”, this work that was previous published in 2018 reach similar conclusions the only variant is the thuringiensis strain and the use of other bacteria termed Serratia liquefaciens. Therefore, the originality of the present study with respect to the other in terms of the findings it refers, and their usefulness is not clear, since when reading both it seems that the conclusions are the same.

Response: I have already described the article “Metal-immobilizing Serratia liquefaciens CL-1 and Bacillus thuringiensis X30 increase biomass and reduce heavy metal accumulation of radish under field conditions” in the introduction. This paper mainly describes the effects of two strains on the growth and accumulation of heavy metals in radish. However, our study focused on the effects of bacteria combined with biochar on the growth and accumulation of heavy metals in radish. The main innovations of our paper are as follows: 1. Strain HC-2 significantly increase the concentration of NH4+ in solution, thereby increasing the pH of solution. 2. Strain HC-2 combined with biochar has a higher effect of inhibiting the enrichment of heavy metals in radish than that of strain HC-2 alone or biochar alone. 3. Strains HC-2, BC and BC+ HC-2 increased the NH4+-N and NO3--N contents, and the NH4+/ NO3- ratio of the radish rhizosphere soils.

It is not understood why to include in the work development a group in Figure 2 where the existence of the thuringiensis strain designated as x-30 is mentioned, which was evaluated in the previous work published in 2018. It is important to mention that This strain was never described or mentioned in the M and M section where only strain HC-2 is included. So, it is not clear, why a result was included in the figures of a strain that was not evaluated?

Response: This is my mistake. The X30 in Figure 2 should be replaced by BC-2. I have revised it in the text.

In Figure 1 (page 6.7) there is no statistical analysis, so it is not possible to establish differences or be able to draw a conclusion of the results.

Response: Thank you for your suggestion. I had already added statistical analysis in Figure 1.

On page 9 L232 the P value is P<0.05 on the other hand in the foot note of Figure 3 the p value mentioned is P>0.05 pleas homogenize this change can led to confusion review in all figures.

Response: This problem has been corrected.

In Table 1 lack of statistical analysis was observer, moreover, the strain HC-2 is in all groups including the control. How is possible to found differences between groups?

Response: Thank you for your suggestion. I had already added statistical analysis in Table 1. Strains BC-2 were analyzed for its colonization in the rhizosphere soils of the radish. Through 16S rRNA gene sequencing and biochemical characterization, the inoculated bacteria were viable in the rhizosphere soils of the radish plants after inoculation for 42 days. Notably, the proportion of HC-2  in the group with strain HC-2 (31.6%) and BC+ HC-2(43.3%) were higher than the proportion of HC-2 in the control group (9.6%) and BC (11.2%).

Please discuss the role of strain HC-2 as endophytic bacteria.

Response: It's a very interesting question. Rhizosphere strain HC-2 was cultured from rhizosphere soils of amaranth. In this study, it was used as a rhizosphere bacteria to immobilize heavy metals, thereby reducing the absorption of heavy metals by plants. Of course, It may also act as a potential endophyte, affecting plant growth and heavy metal uptake, which is the research we will carry out in the follow-up experiments.

Under the stress of heavy metals, some bacteria enter and cooperate with host plants to resist the stress of heavy metals. Endophytes have many excellent characteristics, such as the absorption and tolerance of heavy metals. The advantages of endophytic bacteria have attracted the attention of researchers [37,38]. Studies have shown that plant growth-promoting bacteria can reduce the toxicity of heavy metals to plants, secrete substances that promote plant growth, and change the uptake and transport of heavy metals by plants [13,37,38]. Therefore, plant growth is promoted after plants have been inoculated with plant growth promoting bacteria. In addition, the efficiency of heavy metal damage repair by plants themselves has been improved. Therefore, compared with chemical regulation, PGPB have greater potential and advantages in regulating plant growth because PGPB regulation relases no pollution into the environment and is low risk, very cheap, and more closely related to plants.

Is important to discuss in more detail the relationship between the B. thuringienis proteins and the heavy metal absorption.

Response: Thank you for your suggestion. I had discussed the the relationship between the B. thuringienis proteins and the heavy metal absorption in the text.

Heavy metal ions interact with chemical groups on microbial surface proteins, polysaccharides and lipids to form metal complexes, which are adsorbed and fixed on cell surfaces. Extracellular polymeric substances (EPS), mainly composed of protein and polysaccharides,are the main components of sludge organics. EPS traditionally exhibite a porous polymeric structure,and polar heavy metals can be easily adsorbed onto adsorption sites (by their functional groups such as carboxyl and hydroxyl groups ) on the surface of EPS. Overall,the physicochemical structural characteristics of EPS significantly affect the adsorption processes of heavy metals. Sheng et al. [28] demonstrated that the oxygen atoms in the carboxyl groups of proteins contributed to the adsorption removal of Cu2+. A previous study by Wei et al. [29] found that the complexation of Zn(II) with O–H and N–H groups and C-O stretching significantly decreased the fluorophore intensity of protein- and humic-like substances. In our study, strain HC-2 significantly (P<0.05) decreased the concentrations of Cd and Pb in solution.

Please carefully review the spelling and English along the document in order to avoid mistakes i.e L 275 pag 1” in the group of contral” should be “In the control group”

Response: This problem has been corrected.

I strongly recommended the article need to be reviewed by a statistical expert and a English native speaker.

Response: The revised manuscript has been edited by native English speaker at American Journal Experts and statistical expert.

Round 2

Reviewer 3 Report

In my opinion, the manuscript can be published in present form.